# Possible Impact of Peripheral Inflammatory Factors and Interleukin-1β (IL-1β) on Cognitive Functioning in Progressive Supranuclear Palsy–Richardson Syndrome (PSP-RS) and Progressive Supranuclear Palsy–Predominant Parkinsonism (PSP-P)

**DOI:** 10.3390/ijms252313211

**Published:** 2024-12-09

**Authors:** Patryk Chunowski, Dagmara Otto-Ślusarczyk, Karolina Duszyńska-Wąs, Agnieszka Drzewińska, Andrzej Załęski, Natalia Madetko-Alster, Alicja Wiercińska-Drapało, Marta Struga, Piotr Alster

**Affiliations:** 1Department of Neurology, Medical University of Warsaw, 03-242 Warsaw, Poland; patryk.chunowski@wum.edu.pl (P.C.); karolina.duszynska@gmail.com (K.D.-W.); agnieszka.drzewinska@gmail.com (A.D.); natalia.madetko@wum.edu.pl (N.M.-A.); 2Department of Biochemistry, Medical University of Warsaw, Banacha 1, 02-097 Warsaw, Poland; dagmara.otto@wum.edu.pl (D.O.-Ś.); marta.struga@wum.edu.pl (M.S.); 3Department of Infectious and Tropical Diseases and Hepatology, Medical University of Warsaw, Wolska 37, 01-201 Warsaw, Poland; andrzej.zaleski@wum.edu.pl (A.Z.); alicja.wiercinska-drapalo@wum.edu.pl (A.W.-D.)

**Keywords:** interleukin-1β (IL-1β), interleukin-6 (IL-6), peripheral inflammatory markers, Montreal Cognitive Assessment (MoCA), Frontal Assessment Battery (FAB), progressive supranuclear palsy (PSP)

## Abstract

Progressive supranuclear palsy (PSP) is a tauopathic atypical parkinsonian syndrome. Recent studies suggest that inflammation may play a role in PSP pathogenesis, highlighting markers like the neutrophil-to-lymphocyte ratio (NLR), platelet-to-lymphocyte ratio (PLR), and cytokines such as IL-1β and IL-6. This study aimed to assess the relationship between peripheral inflammatory markers and psychological abnormalities in PSP-RS and PSP-P patients. The study included 24 participants: 12 with PSP-RS, 12 with PSP-P, and 12 controls. Cognitive function was assessed using the Montreal Cognitive Assessment (MoCA); however, the executive functions were evaluated using the Frontal Assessment Battery (FAB), while inflammatory markers such as IL-1β, IL6, NLR, and PLR were measured. The parameter correlation was executed using Spearman’s correlation (rs). The analysis revealed significant negative correlations between NLR and MoCA (rs = −0.48), as well as between PLR and MoCA (rs = −0.60). The negative correlation between IL-1β and MoCA was statistically significant but relatively weak. This study highlights the relevance of inflammatory markers such as NLR and PLR in reflecting cognitive decline in PSP patients, with IL-1β potentially playing a protective role in cognitive function.

## 1. Introduction

Progressive supranuclear palsy (PSP) belongs to atypical parkinsonisms; generally, it is characterized by bradykinesia, impaired postural reflexes, eye movement disturbances, and language or cognitive difficulties [1,2]. Among the hypotheses attempting to explain the pathophysiological mechanism of PSP are the vascular, environmental [3,4,5], genetic [6], and increasingly popular inflammatory mechanisms. PSP is a four-repeat (4R) tauopathy. Neuropathologically, the diagnosis of PSP relies on identifying neurofibrillary tangles and threads within subcortical nuclei, along with the presence of tufted astrocytes [7]. The definitive diagnosis of PSP can be established based on neuropathological examination. Without the possibility of neuropathological confirmation, a diagnosis of probable PSP can be made based on the gradual progression of symptoms, sporadic occurrence, and an onset after the age of 40 [2], but the last two criteria are questioned, especially in relation to the genetic form of PSP [6]. This tau pathology leads to the development of several clinical phenotypes, with the two most common being PSP–Richardson’s syndrome (PSP-RS) and PSP with predominant parkinsonism (PSP-P). Together, these two subtypes account for 80–90% of all PSP cases [8]. According to the most contemporary diagnostic criteria, both PSP types are diagnosed based on an onset of symptoms at age 40 or older, gradual progression of symptoms, and sporadic occurrence [2]. The diagnosis of PSP-RS is established on the basis of vertical eye movement and the tendency to fall within 3 years. In PSP-P, the criteria indicate a marked severity of symptoms commonly linked to parkinsonism, such as freezing and rigidity, which could be accompanied by resistance to levodopa treatment. The diagnostic criteria for PSP include the assessment of cognitive dysfunction [2]. There are various manifestations of cognitive deterioration in PSP, among which mild deficits ranging to pronounced dementia could be mentioned [9]. It comprises a broad range of cognitive abnormalities, including impairments in attention, executive functioning, learning, and memory [10]. In the early stages of the disease, mild cognitive impairment (MCI) is the most prevalent type of cognitive dysfunction, affecting 43% of patients, followed by dementia in 41%, and normal cognition in 16% of subjects is observed. Over time, most PSP patients advance to dementia with an incidence rate of 241 per 1000 patients per year [9]. However, PSP-RS also affects the executive functions, which are related to but not synonymous with frontal lobe function. Research has shown deficits in short-term memory and spatial working memory, along with poor memory strategies [11]. The research indicates that cognitive decline among PSP patients is linked to microscopic indicators of the disease, such as the accumulation of tau [12]. PSP individuals present brain atrophy, which is more pronounced in the frontal lobe [13], hippocampus [14], insula [15], or thalamus [16]. SPECT examination also showed a reduced perfusion among the patients with elevated HbA1C levels in these regions [17]. The neutrophil-to-lymphocyte ratio (NLR), platelet-to-lymphocyte ratio (PLR), and neutrophil high-density lipoprotein ratio (NHR) [18] as well as cytokines released by activated microglia, including TNF-α, IL-1β, and IL-6, are linked to neurodegeneration including atypical parkinsonism disorders like PSP [19]. However, the mechanisms underlying the development of PSP are still not fully recognized [20]. Microglia might be activated by environmental toxins, cytokines, neuronal damage [21], and β-amyloid or tau [22]. The objective of the study was to investigate the association between readily accessible peripheral inflammatory markers, including NLR, PLR, IL-1, and IL-6, and cognitive decline as measured by the widely available MoCA screening tool. Furthermore, the relationship between these inflammatory markers and executive function impairments was examined using the FAB, another standardized screening measure. The study was conducted on cohorts with PSP-RS and PSP-P, with comparisons drawn against a healthy control group.

## 2. Results

Correlation analysis:

A statistically significant negative correlation was found between NLR and MoCA (rs = −0.48, *p* value < 0.009), PLR and MoCA (rs = −0.60, *p* value < 0.001), and NLR and FAB (rs = −0.38, *p* value < 0.03). We also found a positive correlation between IL-1β and MoCA (rs = 0.38, *p* value < 0.03). These results are presented in Figure 1, Figure 2, Figure 3 and Figure 4. The correlations between IL-6 along with other measured parameters and either MoCA or FAB revealed insignificant findings.

**Figure 1 ijms-25-13211-f001:**
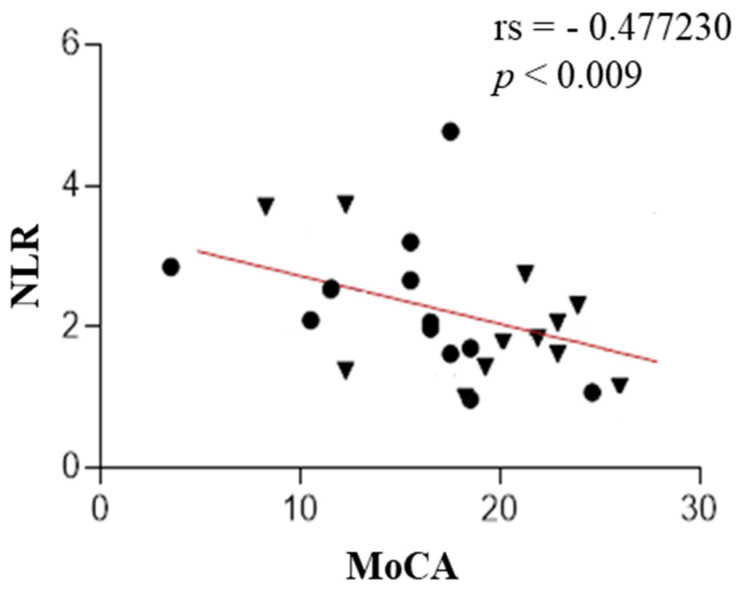
The correlation analysis between the neutrophil-to-lymphocyte ratio (NLR) and test MoCA in patients with PSP (the PSP-RS and PSP-P groups are combined). CirclPSP-RS. Triangle—PSP-P.

Scatter plots show that the NLR was negatively correlated with test MoCA.

**Figure 2 ijms-25-13211-f002:**
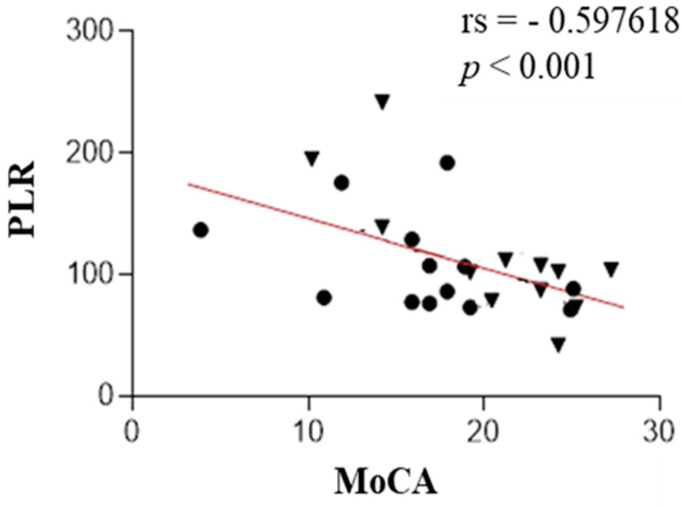
The correlation analysis between the platelet–lymphocyte ratio (PLR) and test MoCA in patients with PSP (the PSP-RS and PSP-P groups are combined). Circle—PSP-RS. Triangle—PSP-P.

Scatter plots show that the PLR was negatively correlated with test MoCA.

**Figure 3 ijms-25-13211-f003:**
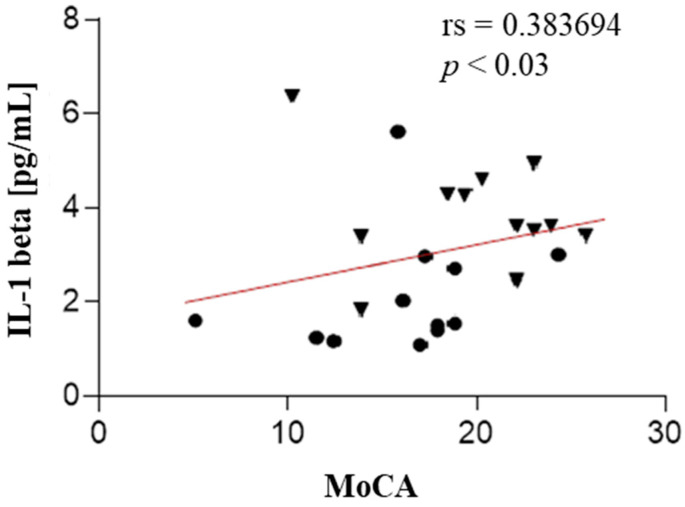
The correlation analysis between the serum IL-1 beta and test MoCA in patients with PSP (the PSP-RS and PSP-P groups are combined). Circle—PSP-RS. Triangle—PSP-P.

Scatter plots show that the serum IL-1 beta was positively correlated with test MoCA.

**Figure 4 ijms-25-13211-f004:**
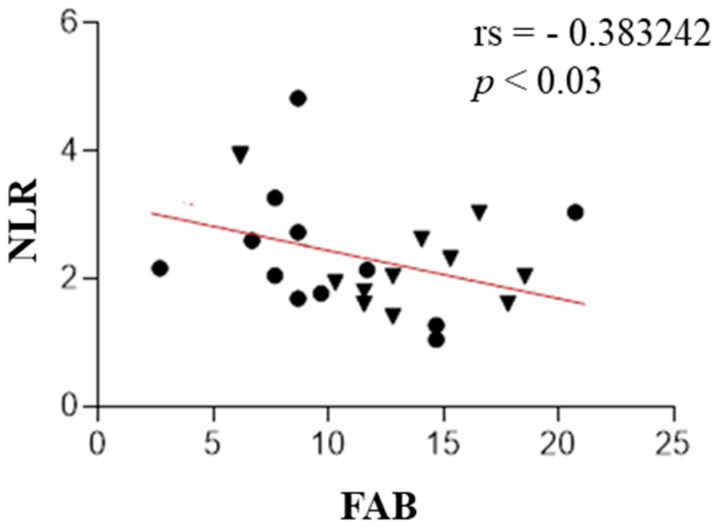
The correlation analysis between the neutrophil-to-lymphocyte ratio (NLR) and test FAB in patients with PSP (the PSP-RS and PSP-P groups are combined). Circle—PSP-RS. Triangle—PSP-P.

Scatter plots show that the NLR was negatively correlated with test FAB.

## 3. Discussion

In this research, the authors evaluated the correlation of IL-1-beta concentration, IL-6, NLR, and PLR with cognitive function assessed in screening, easily accessible tests—MoCA and FAB. IL-1 and IL-6, which are microglial-derived, share common functions in the body’s inflammatory response. IL-1 acts as a leukocytic pyrogen, a mediator of fever, an endogenous leukocytic mediator, and an inducer of components in the acute-phase response. Similarly, IL-6 is produced at the onset of inflammation, traveling through the bloodstream to the liver, where it rapidly stimulates the production of numerous acute-phase proteins. Together, both cytokines play crucial roles in initiating and regulating the acute-phase response during inflammation [23,24]. The role of the mentioned interleukins in the neurodegeneration development is unambiguous. Both IL-1β and IL-6 play crucial roles in driving neuroinflammatory processes, serving as possible mediators in the progression of neurodegeneration [25,26,27]. However, their roles differ in certain aspects. IL-1β primarily acts as a potent pro-inflammatory cytokine, intensifying neural damage [27], whereas IL-6 exhibits a dual function, participating not only in inflammation but also in neuronal and cerebrovascular repair mechanisms [25]. Elevated levels of IL-6 have been linked to the early onset of PD [25,26], suggesting its potential as a biomarker and therapeutic target [25]. Hepcidin, which regulates iron metabolism, is also a component of the nonspecific immune system involved in pathogen response, connecting it to the issue of neuroinflammation in the development of neurodegenerative diseases. Hepcidin suggests a potential acceleration of neurodegeneration as a result of disrupted metal homeostasis [28]. Munoz-Delgado et al. conducted a meta-analysis of 121 individuals, concluding that patients with PSP had a higher PLR [29] and NLR in peripheral blood compared to HCs [29,30]. Our study also shows that, widely described in the literature, elevated NLR and PLR parameters in PSP are associated with lower scores on the MoCA test. This research also shows that the elevated NLR parameter in this disease is associated with lower MoCA test scores. Consequently, increased peripheral inflammatory parameters correlate with poorer performance in executive functions (verbal fluency, cognitive flexibility) and memory (short-term verbal memory, retrieval). Additionally, NLR is negatively correlated with the FAB test to a similar degree; interestingly, no such relationship was observed for PLR.

These findings are consistent with the literature, where patients with PSP are reported to have lower scores on MoCA and FAB tests compared to the control group. It is hypothesized that this may be linked, among other factors, to hypometabolism [31] or cortical atrophy [32,33]. It should be noted that both the MoCA and FAB test are the screening tools for assessing executive functions and memory. However, further research should employ more advanced neuropsychological batteries to analyze which areas of executive functions, language functions, and/or memory are impaired due to elevated levels of peripheral inflammatory factors, and IL-1β. Most studies have indicated a general link between microglial activation, IL-1β, IL-6 secretion, and tau pathology [19]. Regarding Alzheimer’s disease (AD), it is postulated that increased levels of activated microglia are linked to reduced rates of amyloid accumulation as well as slower tau deposition, resulting in cognitive decline. Similarly, stimulated microglial macrophages predicted slower tau accumulation with mitigated cognitive decline as a consequence [22]. Thus, it seems that a similar mechanism may occur in the case of PSP, which aligns with the obtained results, showing a positive correlation between IL-1β, which is microglia-derived, and better cognitive or executive functions among PSP patients. However, the exact mechanism behind this correlation remains unclear. A statistical analysis showed that the mean IL-1β concentration in CSF was significantly higher in patients with PSP-P compared to those with PSP-RS, and in the control group compared to the PSP-RS group [34]. In our study, there is a positive correlation between the IL-1β level and MoCA score; moreover, in the control group, IL-1β concentration is the highest, which may suggest a protective function of interleukins against the development of inflammation. Considering that IL-1β levels are statistically higher in PSP-P than in PSP-RS, and the cognitive function deterioration is more pronounced in PSP-RS, it therefore supports the hypothesis of the protective effect of IL-1β. This may result from the fact that IL-1β, released during the initiation of inflammation, has a self-limiting effect on the microglial cells that secrete interleukins and it secondarily limits inflammation and reduces the decline in cognitive function. Probably due to the correlations obtained are relatively low; nonetheless, a clear negative trend exists between peripheral inflammatory factors and the results of screening psychological tests, along with a positive trend between IL-1β concentration and MoCA scores. No significant correlation was observed between IL-1β concentration and FAB, which may be related to the protective role of interleukins in memory and language functions, without a significant effect on executive functions mainly depending on the frontal lobe. The relationship between IL-6 and the MoCA and FAB tests was also examined in this publication; however, no statistically significant correlation was found between these parameters. Screening tests assessing cognitive functions were used due to their wide availability and ease of use; moreover, the MoCA and the FAB were used due to their broad range of cognitive domains assessed by the tests. As a pilot study, we aimed to use a screening diagnostic tool that evaluates multiple domains to determine whether there is any correlation between cognitive functioning and peripheral inflammatory factors in the most common PSP subtypes.

To our knowledge, no prior studies have examined the relationship between this parameter and peripheral inflammatory factors. However, in the future, more advanced neuropsychological batteries should be employed to identify the specific domains of neuropsychological functioning affected by inflammation. Moreover, both apathy and depression negatively impact on overall cognitive performance [35]. It is estimated that more than half of individuals with PSP exhibit features of depression and/or apathy, although these symptoms are relatively mild in intensity. Major Depressive Disorders are relatively rare. However, there are no adequate tools for making a definitive diagnosis, and much of the data on affective disorders in PSP rely on subjective responses from participants. Nevertheless, even mild affective disturbances can slightly affect the results of tests assessing cognitive functions [35,36]. This implies the need to study these factors using appropriately dedicated tests in future research.

Both PSP groups exhibited decreased thickness and volume in the frontal lobe regions; however, PSP-P showed more extensive cortical thinning, also affecting the temporal and parietal lobes in comparison to PSP-RS [37]. Greater cortical atrophy (especially in the frontal cortex) among PSP-P patients, along with less pronounced cognitive deterioration compared to PSP-RS, strongly supports an inflammatory etiology for the reduction in the cognitive reserve in PSP. However, the studies are not congruent in this field. Another study claims that there is a greater volume loss in the frontal lobe in PSP-RS compared to PSP-P [38]. In other imaging examinations such as SPECT and FDG PET, frontal lobe hypoperfusion and frontal lobe hypometabolism might be observed, respectively, regarding PSP [39,40]. In SPECT, the D2 receptor radiotracer proved to be useful for distinguishing PSP-RS from PSP-P, as striatal uptake was decreased in PSP-RS and slightly elevated in PSP-P. However, there is insufficient data to definitively differentiate PSP-RS from PSP-P based on frontal lobe hypoperfusion [25] or frontal lobe hypometabolism [40]. According to Black and colleagues, frontal hypometabolism is not useful for distinguishing between PSP-RS and PSP-P [31]. The correlation of the obtained results with imaging examination would allow for an assessment of the extent to which cognitive impairments in PSP are caused by general brain atrophy, particularly in the frontal lobe region, versus the contribution of inflammation, most likely triggered by excessive microglial activation in response to the presence of the neurodegenerative tau. Another important marker of PSP is tau quantitative assessment, reflecting the neurodegenerative nature of the disease. The average regional tau severity was higher in PSP-RS than in PSP-P across all brain regions, with a significant difference observed in every region. The PSP-RS group showed a significantly higher total tau concentration compared to the PSP-P group [41]. A higher concentration of tau is most likely directly proportional to the degree of CNS degeneration; therefore, it seems reasonable to conclude that the level of neurodegeneration is positively correlated with cognitive impairment. While PET has a limited functionality in differentiating PSP-RS and PSP-P based on frontal lobe hypometabolism, PET radiotracers bind tau effectively. The inclusion of tau PET significantly increased confidence in determining the underlying etiology. The largest effect sizes for the certainty of etiology and diagnosis were observed in the Aβ-positive group [42]. The 11C-PK11195 radiotracer (TSPO) receptor is rapidly identified as over-expressed in brain lesions and is considered a useful marker for neurodegeneration [43] such as cognitive impairments. The combined use of C-labeled Pittsburgh Compound B ([11C]PiB) and 18F-labeled AV-1451 enables researchers to assess β-amyloid and tau accumulation [44]. In PSP, increased binding has been observed in the basal ganglia, midbrain, frontal lobe, and cerebellum and in the putamen, thalamus, and pallidum. The radiotracer binding in the pallidum, midbrain, and pons has been associated with disease severity [45]. Furthermore, both mentioned radiotracers are linked to microglial activation [44]. PET molecular imaging of microglial activation and tau pathology can potentially forecast clinical progression in PSP, but the exact mechanism of this connection remains unclear.

This study has several limitations. Neuropathological evaluation was not conducted in the study as all patients remain alive. The study is based on possible and probable diagnoses of PSP. Due to the pilot character of the study, no neuroimaging was conducted; however, the authors plan to extend the evaluation in future works. The results of both the MoCA and FAB tests are influenced not only by inflammatory factors but also by years of education, intelligence, affect presented during the examination, and the presence of other conditions. A methodological limitation in this study is that both tests were conducted only once, so the patient’s well-being on the day of testing may have also affected the results. Additionally, the study group was relatively small (24 subjects and 12 subjects in the healthy control group). Study participants were not divided into PSP-RS and PSP-P subgroups, which might slightly alter final results.

## 4. Materials and Methods

### 4.1. Study Group

The research group consisted of 36 individuals: 12 with PSP-RS, 12 with PSP-P, and 12 healthy individuals as the HC. Patients enrolled in the study had to meet the following criteria: provide written informed consent to participate in the clinical trial, be over 55 years of age, show no clinical signs of infection during the internal medicine examination, and have no prior diagnosis of any psychiatric disorder before the PSP diagnosis. Eligible patients diagnosed with either PSP-RS or PSP-P fulfilled the MDS criteria for PSP [2]. The disease duration ranged from 3 to 6 years. In the first group, there were 5 females and 7 males, and in the second group, there were 4 females and 8 males. The mean age of PSP-RS and PSP-P patients was similar (70.3 vs. 68.8); HC was age- and sex-matched. HC had to meet the same criteria as the study group, except for the diagnosis of PSP. Both research groups were separately compared to the control group. Data concerning the individuals participating in the study are summarized in Table 1.

### 4.2. Montreal Cognitive Assessment (MoCA)

The Montreal Cognitive Assessment (MoCA) provides an overall score along with six index scores for specific cognitive domains, which include language, attention, orientation, memory, executive functioning, and visuospatial ones [46]. The MoCA is a screening 10 min paper-and-pencil test with a maximum score of 30 points, where a score below 26 indicates cognitive deficits [47]. MoCA was used.

### 4.3. Frontal Assessment Battery (FAB)

The Frontal Assessment Battery (FAB) test effectively detects frontal lobe dysfunction in PSP [31]. It provides an overall score along with six index scores for specific cognitive domains, which include mental flexibility, conceptualization, environmental autonomy, inhibitory control, programming, and sensitivity to interference [48]. A limitation of this scale is its incomplete assessment of cognitive functioning [49].

### 4.4. Patient Materials

Blood and cerebrospinal fluid (CSF) samples were collected from 24 patients with progressive supranuclear palsy (PSP) hospitalized in the Department of Neurology at the Medical University of Warsaw. The control group comprised 12 healthy individuals who were admitted to the Department of Infectious Diseases, Tropical Diseases, and Hepatology at the same university. These individuals did not have any comorbidities likely impacting the level of inflammatory factors. Blood samples (5 mL) were collected in tubes without anticoagulants, centrifuged, and stored at −80 °C. Similarly, CSF (10 mL) was obtained through a lumbar puncture, frozen, and kept at −80 °C until the analysis.

### 4.5. Biochemical Analysis

Commercial enzyme-linked immunosorbent assays (ELISAs) were employed to measure the levels of IL-6 and IL-1β. Human IL-6 HS and IL-1β ELISA kits (Diaclone SAS, Besancon, France) were used for this purpose. The absorbance was recorded at 450 nm with a plate reader, and concentrations of the markers were determined based on standard curves.

All patients underwent laboratory testing, which involved a full blood count, an evaluation of C-reactive protein (CRP) levels, and biochemical tests, including lipid profiling and ferritin measurement. None of the patients had elevated inflammatory markers such as CRP or leukocytosis. The ratios including the neutrophil-to-lymphocyte ratio (NLR), lymphocyte-to-monocyte ratio (LMR), neutrophil-to-HDL-C ratio (NHR), platelet-to-lymphocyte ratio (PLR), and neutrophil-to-monocyte ratio (NMR) were calculated by dividing the number of neutrophils by lymphocytes, lymphocytes by monocytes, neutrophils by HDL-C, platelets by lymphocytes, and neutrophils by monocytes, respectively. The Sysmex XT 4000i automatic hematology analyzer, located at the Laboratory Diagnostics Department of the Mazovian Hospital in Brodno, was used for the counts.

### 4.6. Statistical Analysis

A data analysis was performed using GraphPad Prism 8 software (GraphPad Software, San Diego, CA, USA). Arithmetic means (X) and standard deviations (SDs) were computed. Statistical significance was established at *p* < 0.05. The normality of the data was evaluated using the Shapiro–Wilk test. Group comparisons were made using the Mann–Whitney U test, and Spearman’s Rank-Order Correlation was applied to assess the relationships between clinical markers and FAB, as well as MoCA test scores.

## 5. Conclusions

In this research, the authors evaluated the association of IL-1-beta concentration, IL-6, NLR, and PLR with cognitive function assessed in screening, easily accessible tests—MoCA and FAB. Nonspecific inflammatory parameters suggest that this process may play a role in cognitive and executive functioning impairments observed in PSP; however, a further detailed analysis is required to elucidate its significance. Although there is increasing interest in studying neuroinflammation and its connection to immune status in various neurodegenerative disorders, limited information is available on PSP. Gaining a deeper understanding of the factors accelerating PSP progression including cognitive deterioration could ultimately lead to the development of new therapeutic strategies.

## Figures and Tables

**Table 1 ijms-25-13211-t001:** Clinical and biochemical data of the study group.

	PSP-RS Patients (n = 12)	PSP-P Patients (n = 12)	Healthy Control (n = 12)
Age [years]—Mean ± SD	70.3 ± 3.6	68.8 ± 6.7	50 ± 8.8
Sex distribution [female/male]	5/7	4/8	7/5
Average disease duration [years]	3.5	4.5	-
CRP [mg/L]	<5	<5	<5
Average NLR—Mean ± SD	2.4 ± 1.0	2.3 ± 0.8	3.03 ± 1.8
Average PLR—Mean ± SD	112.3 ± 28.1	113.2 ± 45.6	97.6 ± 60.2
Average IL-1β [pg/mL] in serum—Mean ± SD	2.2 ± 0.6	3.9 ± 1.1	1.5 ± 0.9
Average IL-6 [pg/mL] in serum—Mean ± SD	4.4 ± 1.5	7.2 ± 2.6	3.9 ± 1.2

PSP-RS, Progressive Supranuclear Palsy–Richardson Syndrome; PSP-P, Progressive Supranuclear Palsy–Predominant Parkinsonism; NLR, Neutrophil-to-Lymphocyte Ratio; PLR, Platelet–Lymphocyte Ratio; SD, Standard Deviation.

## Data Availability

The original contributions presented in this study are included in the article. Further inquiries can be directed to the corresponding author.

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
