# Peer review of "Possible Impact of Peripheral Inflammatory Factors and Interleukin-1β (IL-1β) on Cognitive Functioning in Progressive Supranuclear Palsy–Richardson Syndrome (PSP-RS) and Progressive Supranuclear Palsy–Predominant Parkinsonism (PSP-P)"

_ijms, 2024, doi:10.3390/ijms252313211_

Round 1
Reviewer 1 Report
Comments and Suggestions for Authors
This study explored the relationship between inflammatory markers and cognitive impairment in Progressive Supranuclear Palsy (PSP), focusing on two subtypes: PSP-Richardson syndrome and PSP-parkinsonism. By analyzing blood inflammatory markers and cognitive tests, researchers found that neutrophil-lymphocyte ratio and platelet-lymphocyte ratio were negatively associated with cognitive performance, while interleukin-1β showed a positive correlation. The findings suggest inflammation's potential role in cognitive decline, with IL-1β potentially having a protective cognitive function. However, the researchers acknowledged the study's limitations, including a small sample size, and recommended further research with larger cohorts and neuroimaging studies to validate these initial results.
1)We request a comprehensive table of demographic and inflammatory markers, detailing patient age, sex distribution, disease onset duration, and key blood markers like IL-6, IL-1β, CRP, NLR, and PLR. A well-constructed table will provide critical insights into the PSP patient cohort's characteristics.
2)Blood biomarker correlation with cognitive assessments (FAB and MoCA) offers a patient-friendly alternative to cerebrospinal fluid analysis. This approach provides minimally invasive sampling, enhanced test repeatability, and broader clinical applicability. It allows for a more nuanced understanding of inflammatory processes and cognitive decline in neurological conditions.
3)We recommend refining graphical representations in Figures 1-3 to distinctly differentiate PSP subtypes. Maintain the current PSP-Richardson syndrome (PSP-RS) marker while introducing a unique symbol—such as triangles or x—for PSP-Parkinsonism (PSP-P). This will improve visual clarity and subtype identification.
4)As a pilot study, this research highlights the need for future neuroimaging investigations. Subsequent studies should explore whether PSP's cognitive impairment stems from inflammatory processes or brain atrophy, focusing particularly on frontal lobe changes. Despite current limited discourse, a more comprehensive exploration is crucial to understanding PSP's pathogenesis.
5)Interleukin-1β (IL-1β)'s potential role in mitigating cognitive decline, especially in PSP-Richardson syndrome, presents a complex scientific puzzle. The contrast between prominent astrocytic changes and minimal microglial alterations raises profound questions about inflammatory dynamics. This paradoxical relationship represents the most intriguing aspect of current PSP research, demanding sophisticated further investigation.
Reviewer 2 Report
Comments and Suggestions for Authors
This article presents interesting findings, but authors need to revise the below points:
The Methods section should be above the Results section.
How was the sample size estimated?
Were participants matched on a one-to-one basis? Please describe in the text.
How were the participants recruited? Please explain.
Authors need to provide a more extended Introduction. For example, mood is not examined, but it is an important variable that may have influenced cognitive performance in the tests and this point should be also added in the Limitations of the study (for a relevant recent article on the importance of depression and apathy as moderators/mediators on these tests scores for this age group to add and discuss: https://www.mdpi.com/2076-3425/11/6/785) and a review highlighting the existence of depression in PSP (https://journals.sagepub.com/doi/abs/10.1177/0891988721993545).
Why were these cognitive tests chosen instead of others? Please justify in the Methods section.
Round 2
Reviewer 2 Report
Comments and Suggestions for Authors
Most points raised by the reviewers have been answered in this revision.